# pH and Pectinase Dual-Responsive Zinc Oxide Core-Shell Nanopesticide: Efficient Control of Sclerotinia Disease and Reduction of Environmental Risks

**DOI:** 10.3390/nano14242022

**Published:** 2024-12-16

**Authors:** Qiongmei Mai, Yu Lu, Qianyu Cai, Jianglong Hu, Yunyou Lv, Yonglan Yang, Liqiang Wang, Yuezhao Zhou, Jie Liu

**Affiliations:** College of Chemistry and Materials Science, Jinan University, Guangzhou 510632, China; 13702739503@163.com (Q.M.); 19809570662@163.com (Y.L.); qycai_dgut@163.com (Q.C.); tlouxx0825@163.com (J.H.); moseslu2008@163.com (Y.L.); yangyonglan0219@gmail.com (Y.Y.); wlqqdldcx@163.com (L.W.); 18875930512@163.com (Y.Z.)

**Keywords:** nanopesticides, pH and enzyme dual response, *Sclerotinia sclerotiorum*, synergistic antimicrobial activity, biosafety

## Abstract

*Sclerotinia sclerotiorum* is one of the fungi that cause plant diseases. It damages plants by secreting large amounts of oxalic acid and cell wall-degrading enzymes. To meet this challenge, we designed a new pH/enzyme dual-responsive nanopesticide Pro@ZnO@Pectin (PZP). This nanopesticide uses zinc oxide (ZnO) as a carrier of prochloraz (Pro) and is encapsulated with pectin. When encountering oxalic acid released by *Sclerotinia sclerotiorum*, the acidic environment promotes the decomposition of ZnO; at the same time, the pectinase produced by *Sclerotinia sclerotiorum* can also decompose the outer pectin layer of PZP, thereby promoting the effective release of the active ingredient. Experimental data showed that PZP was able to achieve an efficient release rate of 57.25% and 68.46% when pectinase was added or under acidic conditions, respectively. In addition, in vitro tests showed that the antifungal effect of PZP was comparable to that of the commercial Pro (Pro SC) on the market, and its efficacy was 1.40 times and 1.32 times that of the Pro original drug (Pro TC), respectively. Crucially, the application of PZP significantly alleviated the detrimental impacts of Pro on wheat development. Soil wetting experiments have proved that PZP primarily remained in the soil, thereby decreasing its likelihood of contaminating water sources and reducing potential risks to non-target organisms. Moreover, PZP improved the foliar wettability of Pro, lowering the contact angle to 75.06°. Residue analyses indicated that PZP did not elevate prochloraz residue levels in tomato fruits compared to conventional applications, indicating that the nanopesticide formulation does not lead to excessive pesticide buildup. In summary, the nanopesticide PZP shows great promise for effectively managing *Sclerotinia sclerotiorum* while minimizing environmental impact.

## 1. Introduction

In the realm of agriculture, pesticides, particularly those of a chemical nature, are indispensable for boosting crop yields and mitigating the prevalence of pests and diseases [1,2,3]. Nevertheless, conventional pesticide applications suffer from alarmingly low efficacy rates, posing significant threats to environmental safety [4,5,6]. Enhancing the functionalities of existing formulations through the integration of nano-carrier technology represents a pivotal advancement. This approach involves enveloping traditional pesticides in nano-carriers, which not only amplifies their water solubility and resistance to photodegradation but also facilitates targeted, responsive release at the precise site of pathogen activity. Such innovations markedly elevate the efficiency of pesticide utilization, amplify the potency in combating pest infestations and disease outbreaks, all while diminishing the overall quantity of pesticides necessary for effective control [7,8,9].

When plants are attacked by pathogens, their normal growth is significantly affected. In order to protect plants from interference by these pathogens, people usually focus on the microenvironment where pathogens live, using this as an entry point to explore the theoretical basis and practical possibility of developing new nanopesticides [6,10,11]. Taking *Sclerotinia sclerotiorum* as an example, this bacteria produces oxalic acid after infecting plants, causing the pH value of the surrounding environment to drop. This acidification not only promotes the enhanced activity of various fungal enzymes, but also inhibits the host plant’s natural defense system and hinders the autophagy process. Therefore, regulating the host environment by secreting oxalic acid is a strategy used by *S. sclerotiorum* to promote its own reproduction [12,13,14]. In addition, this type of fungus also releases enzymes that can break down the plant cell wall structure, thereby softening the tissue and facilitating further expansion of the infection range [15,16,17]. Currently, materials such as zinc oxide, calcium carbonate, and MOF are widely used as effective carriers in response to acidic changes [18,19,20]; Enzymatic reactions often require specific triggers for nanomodification [21,22]. For example, in the process of fighting fungal diseases, pectin or cellulose may be used to coat the surface of nanoparticles so that they can interact with corresponding types of fungal enzymes [23,24]. In their work, a research team from the Zhejiang Provincial Key Laboratory of Crop Pathogens and Pest Biology at Zhejiang University used ZnO and ZiF-8 to construct a core-shell structure to wrap the berberine component, forming a structure called Ber@ZnO-Z of nanopreparations. This design allows for the rapid release of the berberine contained in it when encountering acidic conditions, aiming to effectively control the occurrence and development of tomato bacterial wilt disease [25].

Research has demonstrated that zinc oxide (ZnO) can generate reactive oxygen species (ROS), thereby enhancing cell membrane permeability. It also has the unique ability to disintegrate and liberate encapsulated drugs and Zn^2+^ ions in acidic conditions. Zn^2+^ serves as an essential trace element for plant growth, with zinc oxide being non-toxic to human cells [26,27,28]. Pectin, a heterogeneous polysaccharide found in the cell walls of higher plants, is utilized as a wall material in nanocarriers. Additionally, pectin acts as a thickening agent, increasing the viscosity of the aqueous phase and improving the wettability of the nanosystem on leaves. Pectin is highly biocompatible and safe for human consumption, having been included in the World Health Organization/FAO Food Codex Alimentarius [29,30,31].

Nanopesticides boast notable benefits such as robust stability, adhesive properties, and regulated release, positioning them as a plausible replacement for conventional pesticides [32,33,34]. Nevertheless, their steadfast nature poses challenges, including resistance to degradation in agricultural settings, potentially escalating environmental pollution if the nanomaterials themselves are deleterious. The strong adhesion characteristic, while advantageous for non-edible plant parts like leaves, skins, and fruits, raises health concerns when applied to edible produce, necessitating careful design considerations to mitigate these risks [35,36,37,38,39,40,41].

In this study, we developed a novel nanopesticide, PZP, which consists of a pectin-coated zinc oxide nanocarrier loaded with prochloraz, and features a pH/enzyme dual response mechanism (Figure 1). We conducted a systematical characterization of these nanopesticides and evaluated their efficacy in managing *sclerotinia* disease through various experimental approaches. These included assessing their pH/enzyme responsiveness, conducting biosafety evaluations, testing leaf wettability, analyzing pesticide residue levels, examining soil wetting properties, and performing in vitro antibacterial studies [42,43,44,45,46,47,48,49,50,51,52,53]. Our findings indicate that PZP is capable of effectively preventing and controlling *Sclerotinia sclerotiorum* while balancing the benefits and potential risks associated with nanotechnology, thereby offering a promising strategy for the prevention and control of *Sclerotinia sclerotiorum.*

## 2. Experimental Section

### 2.1. Materials

Zinc acetate was purchased from Shanghai Macklin Biochemical Technology Co., Ltd. (Shanghai, China). Prochloraz was purchased from Shanghai Aladdin Biochemical Technology Co., Ltd. (Shanghai, China). Potato Dextrose Agar (PDA) was purchased from AoBoXing Product, Beijing China. Fluorescein Isothiocyanate (FITC) was purchased from Meryer (Shanghai, China) Chemical Technology Co., Ltd. Tomatoes and wheat seeds were purchased from Nancun Comprehensive Market, Panyu District, Guangzhou City, Guangdong Province, China. Sclerotinia GDMCC 3.681 was purchased from Guangdong Microbial Culture Collection Center, GDMCC. Pectin and pectinase from Shanghai Aladdin Biochemical Technology Co., Ltd.

### 2.2. Synthesis and Characterization of Pro@ZnO@Pectin

#### 2.2.1. Synthesis of ZnO

Add 30 mL of diethylene glycol and 16.5 mg of anhydrous zinc acetate into a three-necked flask and disperse them by ultrasonication. Then put it in an oil bath at 160 °C, condense and reflux, and stop the reaction when the white particles turn into white milky substance. The cooled sample is centrifuged at low temperature (10,000 rpm, 30 min), the precipitate is collected, washed with anhydrous ethanol and ultrapure water 3 times each, and finally the sample is freeze-dried to obtain ZnO nanoparticles with rough surface [42,43].

#### 2.2.2. Synthesis of Pro@ZnO@Pectin

Dissolve 20 mg of prochloraz (Hereinafter referred to as Pro) in 10 mL of anhydrous ethanol, add 10 mg of ZnO powder, stir overnight, add 2 mg/mL of pectin solution to the above suspension while stirring overnight again to synthesize the core-shell nanopesticide Pro@ZnO@Pectin (Hereinafter referred to as PZP), centrifuge the suspension, wash with water 3 times, and freeze-dry the precipitate. Without prochloraz, the nanocarrier ZnO@Pectin was synthesized by the same method (Hereinafter referred to as ZP).

### 2.3. Characterization of Nanomaterials

The material morphology was observed by transmission electron microscopy, the elemental composition was analyzed by X-ray photoelectron spectroscopy, the infrared spectrum of the nanomaterial was obtained by nylon 6700 infrared spectrometer, the loading amount of Pro was obtained by TG and high performance liquid chromatography, the zeta potential of the nanomaterial was obtained by nanoparticle size and zeta potential analyzer (NANO ZS90), and the material was analyzed by UV-visible spectrophotometer (Varian, Palo Alto, CA, USA) and X-ray powder diffractometer (MiniFlex600, Rigaku company, Tokyo, Japan).

### 2.4. Responsive Release of Pro@ZnO@Pectin

At different time intervals, the supernatant (2 mL) was removed by centrifugation, and the absorbance (A) of Pro in the sample tube at 228 nm was measured using UV absorption spectrophotometry. After the measurement, the sample was poured back and redispersed. The morphological changes of the nanoparticles before and after release were observed by TEM. The accumulation of Pro released from the nanocomposite was calculated according to the following equation:(Cumulativerelease ratio (%)) E_p_ = ∑_t=0_^t^ Mt/M0.

Where Ep is the cumulative release rate of prochloraz in Pro@ZnO@Pectin (%); Mt is the cumulative release amount at each sampling time; M0 is the amount of Pro initially added.

### 2.5. In Vivo Response Behavior of Pro@ZnO@Pectin

When preparing the culture medium for the antibacterial plate experiment, add materials of different components (PAP, PZP + pectinase, ZnO + pectin, ZnO + pectin + pectinase), and then add 0.5% BB indicator to obtain a blue PDA culture dish. Place the bacterial cake in the center of the culture dish, take pictures on the 2nd, 3rd, 4th, and 5th days, and record the size of the yellow range. On the fifth day, measure the pH of the bacterial cake with precision pH test paper.

### 2.6. Pro@ZnO@Pectin’s Antibacterial Properties

Place an 8 mm bacterial cake in the middle of the culture medium, and use the sterile water group as a control. Place the above culture medium at 28 °C, observe and record the size and diameter of the bacterial cake from 1 to 6 days, and measure the diameter using the cross measurement method to obtain the corresponding inhibition rate.

### 2.7. Antibacterial Experiment of Filter Paper Using Pro@ZnO@Pectin

Pour PDA into the culture dish, cool it down, and then inoculate the Sclerotinia cake into the culture dish. When the hyphae grow to 2.5 cm, place the sterilized filter paper soaked with different components (200 μg/mL and 50 μg/mL Pro, PZP, PSC) at 1 cm away from the hyphae. Use sterile water as a control. When the hyphae grow to the edge of the control group, record the distance from the remaining filter paper edge to the hyphae and calculate the inhibition rate.

### 2.8. Pro@ZnO@Pectin Induced Morphological Damage to Sclerotinia sclerotiorum

After incubating *S. sclerotiorum* and samples of different components (Pro, PZP, PSC) for 48 h, the hyphae were placed on a glass slide, covered with a coverslip, and the morphology of *S. sclerotiorum* was observed under an upright microscope.

### 2.9. Wheat Plant Experiment

After the wheat was cultivated with clean water for a period of time, the wheat with similar growth was divided into 4 groups. Use a sterile needle to pierce 5 wounds on the rhizome of wheat, then add 1 mL of Sclerotinia mycelium suspension. After 1 day, use a small spray bottle to spray about 2 mL of different preparations (sterile water, 50 μg/mL of Pro, PZP, PSC). Observe the growth of wheat plants.

### 2.10. In Vitro Tomato Experiment

After washing the tomatoes with clean water to remove visible impurities, soak them in 75% alcohol suspension for 30 s, rinse with sterile water, then soak them in 2% sodium hypochlorite for disinfection, and finally rinse them with sterile water. The mycelial suspension was prepared according to the previous method, and the OD600 was measured to be 1.02. Drop the mycelium suspension into the cut tomatoes and culture at 26 °C. After mycelium begins to appear on the tomatoes, add different components (100 μg/mL 20 μL of Pro, PZP, PSC) every day. Use an equal amount of sterile water as a control and record the conditions of tomatoes and *S. sclerotiorum* on days 0, 3, 6, and 9.

### 2.11. Leaf Contact Angle of Pro@ZnO@Pectin

The wettability of the nanoformulations was evaluated by measuring the contact angles of the different components on the leaf blade. The cultivated leaves were cut along the main vein to keep the leaves as flat as possible, and then water and different concentrations of Pro, PZP, and PSC were dropped onto the leaves. After 10 s, the droplet images were taken and the contact angles were recorded. After that, the concentration was fixed at 400 μg/mL, and then water, Pro and PZP were dropped on the leaves, and the contact angle changes of the droplets were recorded from 0 to 7 min.

### 2.12. Biosafety of Pro@ZnO@Pectin

Wheat seeds were soaked in different components (water, 1 mg/mL Pro, PZP, PSC and 2 mg/mL PZP), and then the soaked seeds were placed in a culture dish with moist filter paper on the bottom of the dish, 10 seeds in each group, three copies in a group, and cultured in a suitable place. The corresponding component preparations were sprayed every day. The seed germination rate was recorded until the embryo grew to 1–2 mm. The root length, stem length, fresh weight and dry weight of the seeds were recorded after 7 days.

### 2.13. Soil Mobility of Pro@ZnO@Pectin

The collected soil was air-dried at 25 °C and ground to a diameter of less than 2 mm. The treated soil was placed in a plastic column with cotton at the bottom. The top of the soil was covered with quartz sand and filter paper in turn. Deionized water was passed through the column for 1 h in advance until no water droplets fell, and then 300 μL of 8.5 mg/mL PZP (equivalent to 500 μg of Pro) and 1 mL of 500 μg/mL Pro were added to the middle of the filter paper, followed by elution with deionized water and collection of the leachate. After filtering with a 0.22 μm filter membrane, the concentration of Pro was analyzed by HPLC, where the leaching rate was: leaching (%) = M2/M1 × 100%, where M2 represents the amount of Pro in the leaching solution and M1 represents the amount of Pro initially added.

### 2.14. Residue of Pro@ZnO@Pectin on Tomatoes

Wash the cherry tomatoes with clean water and wipe them clean. Then, drop 300 μL of 8 mg/mL PZP (the amount of Pro is 470 μg) and 470 μL of 1 mg/mL Pro on the cherry tomatoes in batches, dropping a small amount each time and allowing them to dry on the surface of the cherry tomatoes. Then the cherry tomatoes were rinsed and soaked with 20 mL of deionized water, respectively, the eluate was collected, and the concentration of Pro was analyzed by HPLC. Calculate the residual Pro, residual Pro (Q) = M1 − M2, where M1 is the initial amount of Pro and M2 is the amount of Pro in the eluent. After obtaining the Pro residue, the maximum residue S (mg/kg) = Q/m was calculated according to the weight of the cherry tomatoes, where Q is the residue Pro and m is the mass of the cherry tomatoes.

### 2.15. Statistical Methods

The data were obtained from three independent repeated experiments. All values are reported as mean ± SD. For multiple group comparisons, one-way ANOVA and Duncan’s test were performed by Statistical Products and Services Solutions (SPSS). *p* < 0.05 was considered statistically significant.

## 3. Results and Discussion

### 3.1. Synthesis and Characterization of PZP and Its pH-Responsive Release

In this study, we synthesized ZnO and loaded prochloraz via physical adsorption, and then coated pectin with nanoparticles via electrostatic interaction to obtain PZP. As shown in Figure 1A, ZnO with a rough surface can be observed through TEM [54]. From the XRD spectra, it can be seen that the synthesized ZnO have obvious diffraction peaks at 30–40°, which correspond to the standard card of hexagonal wurtzite (JCPDS No. 36-1451). From the reference XRD results [49], the (100), (002), (102), (101), (110), (103), (112), and (201) crystal planes of zinc oxide are basically consistent with the peak values of the experiment in this paper (Appendix A). In addition, the crystal size of ZnO was calculated to be about 23 nm using the Scherrer formula. The ZnO nanoparticles prepared by studying the literature are large particles formed by the aggregation of multiple small ZnO particles with a size less than 25 nm [54]. After being coated with pectin, it forms a smooth spherical shape, and the core-shell structure can be clearly seen in Figure 1B, which preliminarily proves that the nanosystem is successfully prepared. Then from the thermogravimetric analysis of Figure 1C, we can see the proportion of each material. The analysis shows that the drug loading rate of nanopesticide is 19.57%, which is close to the HPLC result. It also proves the successful loading of Pro and the successful encapsulation of pectin. In the UV-visible spectrum, the peaks of ZnO and pectin can be clearly seen in ZnO@Pectin and PZP, while there is also a peak of Pro at 228 nm in PZP (Figure 1E,F). The Cl element 2p peak unique to Pro and the 2p peak of Zn are observed in the XPS spectrum (Figure 1D), which once again demonstrates the successful loading of prochloraz. In the FT-IR spectrum, typical strong vibrations of Zn-O can be observed around 440 cm^−1^, and the bands at 1085 cm^−1^ and 1010 cm^−1^ are characteristic of the vibrations of the COC glycosidic bonds in the pectin backbone, again indicating the successful encapsulation of pectin (Figure 1G). In the Zeta potential diagram, after loading with prochloraz, the zeta potential of the system increased from 4.91 mv to 8.87 mv, and after coating with pectin, the potential dropped to −7.54 mv, indicating that pectin was successfully coated by electrostatic action (Figure 1H). Nitrogen adsorption-desorption experiments showed that the specific surface area of zinc oxide was about 47.25 m^2^ g^−1^, which was large enough to be used to load prochloraz (Figure 1I).

In order to create a good growth environment for itself during its growth, *Sclerotinia sclerotiorum* secretes oxalic acid to lower the pH of the infected parts of the plant. It also secretes cell wall degrading enzymes (including pectinase) to destroy the plant structure and provide itself with a carbon source, thereby promoting its penetration and colonization of the plant. Therefore, a nanosystem that responds to acidic pH and cell wall-degrading enzymes is developed so that it can quickly release drugs at the lesion site to inhibit or kill *S. sclerotiorum*. ZnO has excellent acid response function, and Pectin has pectinase response function. The synthesized nanosystem PZP carries their advantages. In order to study the release behavior of PZP, the release process of Pro in vitro was first analyzed under acidic (pH 5.4) and neutral conditions (pH 7.4). TEM images showed that at pH 7.4, the morphology of PZP did not change significantly. When the pH dropped to 5.4, the morphology of PZP became irregular (Figure 1J,K), indicating that PZP responded to the acidic microenvironment. At the same time, the data showed that PZP released Pro at pH 7.4 and pH 5.4, with cumulative release rates of 23.60% and 68.46% after 36 h, respectively (Figure 1L). The Pro release efficiency increased significantly with the decrease in pH. Combined with the microenvironment created by the growth of *Sclerotinia sclerotiorum*, the PZP nanosystem can achieve pH-responsive controlled release of its disease targets.

On the other hand, the release behavior of Pro was analyzed in a solution containing pectinase [55,56]. According to TEM images, in the presence of pectinase, the surface of the nanoformulation was no longer smooth, similar to ZnO, indicating that the pectin on the surface of PZP had been degraded and Pro was released (Figure 1M–O), with the highest cumulative release reaching 57.25%. This indicates that PZP can also respond to the cell wall-degrading enzymes released by *Sclerotinia sclerotiorum* to achieve targeted controlled release [57,58,59].

### 3.2. PZP Nanopesticide Release in Response to the Fungus Sclerotinia sclerotiorum

The high efficiency of PZP in inhibiting *Sclerotinia sclerotiorum* is due to its excellent stimulus-responsive behavior in vitro. The responsive release of PZP in *Sclerotinia sclerotiorum* was verified by plate experiments, using bromophenol blue sodium salt as a pH indicator (indication range 3.0–4.6), corresponding to a color change from yellow to blue. The results showed that after 120 h of incubation, the *S. sclerotiorum* in the Control group had grown all over the culture medium, and its color changed from the initial blue to pure yellow. The growth of *S. sclerotiorum* in the PZP group was inhibited, and its yellow range was reduced, indicating that PZP responded to the microenvironment of *S. sclerotiorum* releasing oxalic acid and released Pro to inhibit the growth of *S. sclerotiorum*. On this basis, after adding pectinase to the PZP + Pectinase group, the range of yellow was further reduced, indicating that PZP responded to pectinase while responding to oxalic acid. The double response resulted in further release of Pro, enhancing the inhibitory effect on *Sclerotinia sclerotiorum*. The carrier ZP group without Pro had almost no antibacterial effect, and after the addition of pectinase, there was still no antibacterial effect (Figure 2A), indicating that PZP showed an obvious antibacterial effect due to the release of Pro in the microenvironment in response to oxalic acid and pectinase. Next, the pH of the Sclerotinia cakes was tested using precision pH test paper. The pH of the cakes without Sclerotinia growth showed blue, the cakes with Sclerotinia growth, and the cakes of the ZP group and the ZP + Pectinase group all showed yellow, while the cakes of the PZP group and the PZP + Pectinase group were between blue and yellow, further confirming the above conclusion (Figure 2B). Figure 2C is a fine pH standard card.

### 3.3. Fungicidal Activity and Mechanism of PZP Nanopesticide

The bactericidal activity of PZP, Pro and PSC was studied using the mycelial growth rate method. The results showed that different concentrations of PZP, Pro, and PSC showed significant fungicidal activity compared with the Control group, and the antibacterial activities of PZP, Pro, and PSC showed a concentration-dependent relationship. At the same time, the antibacterial activities of PZP and PSC were higher than those of Pro at the same concentration (Figure 3A). The data showed that at the same concentration (the equivalent of prochloraz was 1 μg/mL), the colony diameters of PZP and PSC were similar after 144 h of co-incubation (Figure 3B), and their inhibition rates were 89.38% and 84.45%, respectively (Figure 3C); the antibacterial rate of Pro in the control group was 64.02%. At the same prochloraz dose, the daily colony growth of the PZP group was much less than that of the PSC and Pro groups (Figure 3D), indicating that the targeted release of PZP is beneficial to improving the bioavailability of the fungicide. Compared with PMBD mentioned above [42], at high concentrations (the equivalent of prochloraz is 1 μg/mL), PMBD has an antibacterial rate of up to 91.86%, which is better than PZP. This may be due to the synergistic antibacterial effect of MPDA. However, the dual-response release of PZP is completely targeted at the microenvironment of *Sclerotinia sclerotiorum*, which makes the antibacterial rate of PZP higher than that of PMBD at low concentrations (the equivalent of prochloraz is 0.25 μg/mL) at different time periods. At 144 h, the antibacterial rate of PZP is 60.51%, which is 1.44 times higher than the antibacterial rate of PMBD (42.07%).Therefore, both nanosystems have their advantages, and different nanosystems can be selected according to needs to achieve better antibacterial effects. The above conclusions were further verified by the filter paper antibacterial method. The results showed that when the component concentration was 200 μg/mL, the growth of *S. sclerotiorum* was approximately quadrilateral, and its growth was significantly inhibited in the PZP, Pro, and PSC directions (Figure 3E). After the component concentration was reduced to 50 μg/mL, the inhibitory effect weakened and the growth of *S. sclerotiorum* was approximately circular (Figure 3F). At the same time, at 200 μg/mL, the inhibition rates of PZP, Pro, and PSC were 68.0%, 54.0%, and 62.6%, respectively; at 50 μg/mL, the inhibition rates were 31.0%, 18%, and 29.3%, further verifying the above conclusion (Figure 3G).

In order to study how PZP works as an antibacterial agent, the morphology of *S. sclerotiorum* treated with different components (sterile water, Pro, PZP, and PSC) was observed using an upright microscope. The results showed that the hyphae in the Control group were smooth, thick and tubular in shape (Figure 3H), and they grew completely, in large numbers, and in an interlaced growth state (Figure 3L). However, the mycelium treated with Pro, PZP, and PSC showed signs of breakage and growth inhibition (Figure 3I–K), and growth was sparse, mostly in the form of scattered short chains (Figure 3M–O).

### 3.4. Antibacterial Effect of PZP on Wheat Plants and Detached Fruits Infected with Sclerotinia sclerotiorum

Compared with in vitro antibacterial experiments, plants do not completely absorb nanopreparations. When living plants are infected with *Sclerotinia sclerotiorum*, they need to be sprayed with drugs to achieve antibacterial effects [60,61,62,63]. In order to further verify the inhibitory effect of PZP on *Sclerotinia sclerotiorum* in living plants, *Sclerotinia sclerotiorum* was inoculated on living wheat plants, and then Pro, PSC, and PZP were sprayed, and the growth of wheat was observed. The results showed that the wheat in the control group, which did not receive any treatment, began to wilt and fall over on the 8th day. On the 14th day, the wheat was almost completely withered and the leaves turned yellow. In contrast, the wheat in the Pro group showed an antibacterial effect and inhibited the growth of *Sclerotinia sclerotiorum*. There was no large-scale lodging and yellowing of the wheat, and only a small part of the wheat had yellowing leaves, which may be due to the impact of Pro itself on the growth of wheat. The wheat in the PZP and PSC groups not only showed excellent antibacterial effects but also grew vigorously, indicating that the antibacterial effects of PZP and PSC were better than those of Pro (Figure 4A).

Fruit is one of the main sites of infection by *Sclerotinia sclerotiorum*. Therefore, an in vitro fruit experiment was conducted to further evaluate the preventive effect of PMBD on cherry tomatoes infected with *Sclerotinia sclerotiorum*. The results showed that white hyphae began to appear in the tomatoes of each group on the third day after inoculation with *S. sclerotiorum*, indicating that *S. sclerotiorum* had infected them (Figure 4C). At this time, different components were added. On the sixth day, the tomatoes in the Control group were almost covered with hyphae, and most of the tomatoes in the Pro group were also infected. This may be due to the decomposition of Pro, which caused Pro to become ineffective and had no antibacterial effect. However, the tomatoes in the PSC and PZP groups had only some free Sclerotinia hyphae, indicating that PSC and PZP can inhibit the growth of Sclerotinia in tomatoes. By the ninth day, the tomatoes in the Control and Pro groups were almost completely covered by *Sclerotinia sclerotiorum*, and the complete hyphal structure could be seen, while some tomatoes in the PZP and PSC groups were also infected, indicating that the control of PZP and PSC on in vitro tomato fruits has a certain timeliness, and control needs to be carried out within a reasonable time to achieve the desired effect.

When applying pesticides, the biosafety of nanocarriers needs to be considered. We evaluated the safety of PZP and ZnO carriers by soaking wheat seeds in water, 400 μg/mL ZnO, PZP, Pro, and 2 mg/mL PZP (hereafter referred to as PZP 2). The results showed that 400 μg/mL of Pro had a certain inhibitory effect on the germination and growth of wheat. At this concentration, the PZP and ZnO groups had almost no effect on the growth and germination of wheat. When the concentrations of carrier ZnO and PZP were increased to 2 mg/mL (Pro content was about 400 μg/mL), the inhibitory effect of PZP on wheat growth was significantly lower than that of the 400 μg/mL Pro group (Figure 4B). This inhibitory effect indicates that the nanocarrier has weakened the toxicity of Pro to a certain extent. It can also be found from the germination rate that the germination rates of the Water, PZP, and ZnO groups were all above 95%, the germination rate of the PZP 2 group dropped to 90%, and the germination rate of the Pro group dropped to 80% (Figure 4D). In addition, the root and stem lengths of the Water, ZnO, PZP, PZP 2, and Pro groups were 10.31, 9.71, 9.56, 7.81, and 1.94 cm and 8.27, 7.80, 7.76, 6.82, and 4.69 cm, respectively, which was consistent with the above conclusions (Figure 4E,F). This also affected the fresh weight/dry weight, resulting in different degrees of reduction in the fresh/dry weight of the PZP 2 and Pro groups (Figure 4G).

### 3.5. Wettability, Residue on Tomato Surface and Soil Mobility of PZP

The impact of PZP on the environment can be assessed by evaluating the pesticide mobility of PZP in soil and the pesticide residues on fruits. When spraying pesticides, it is difficult for fungicides to reach the target due to splashing and bouncing, or the amount of pesticides reaching the target is extremely small, resulting in the actual effective utilization rate of pesticides being less than 0.1%. Most of the fungicides splashed into the soil will cause secondary pollution. The use of nanocarriers can effectively increase the wettability of fungicides on leaves, reduce the splashing and bouncing of nanopesticides, and improve the effective utilization rate of pesticides and fungicides. PZP benefits from the hydrogen bonds formed between the outer pectin and the fatty acids of the leaves, which effectively increases the deposition of the fungicide on the leaves. The results showed that the contact angle of the PZP group was significantly smaller than that of the Pro group and the Control group (Figure 5A). Moreover, over time, the contact angles of the Control group and Pro group decreased from 119.71° and 119.94° to 105.87° and 110.94° (7 min), respectively, while that of the PZP group decreased from 75.06° to 57.83° (Figure 5D), proving that PZP spreads better on the leaf surface over time. In addition, the comparison of the contact angles of PZP, Pro, and ZnO@Pectin at different concentrations confirmed that the ZnO@Pectin nanocarrier could effectively increase the wettability of the fungicide Pro on the leaves (Figure 5B), and as the concentration increased, the contact angle of the PZP group decreased from 94.62° to 74.46°, and that of the ZnO@Pectin group also decreased significantly (Figure 5C). Compared with PMBD mentioned above, under the same conditions, the contact angle of PMBD is 65.75°, which is smaller than that of PZP, indicating that PMBD has better wettability. For vegetables or fruit crops with edible skin, pesticide residues after application of pesticides need to be considered, and appropriate concentrations and nanocarriers with different wettability should be appropriately selected. In summary, when spraying nanopesticides, it is necessary to rationally select nanocarriers with different wettability according to the type of plant, which is beneficial to improve the utilization of pesticides and will not cause harm to the human body.

In order to evaluate whether PZP would penetrate the soil and cause secondary pollution to groundwater and rivers, a soil column leaching experiment was used to simulate the leaching and migration ability of PZP in the soil (Figure 5E). As shown in Figure 5H, the device was used to measure the Pro content in the leachate, and the retention rate of Pro in the soil was calculated. The results showed that after elution with 50 mL of water, the leaching rate of Pro in the PZP group was 44.41%, while the leaching rate of Pro in the Pro group was 87.53% (Figure 5G), indicating that ZnO@Pectin can reduce the mobility of fungicides in the soil, allowing them to be retained more in the soil, and reducing the pollution of Pro to rivers and groundwater. At the same time, the maximum flux diagram also shows that after elution with different volumes of water, the maximum flux of Pro in the PZP group in the leachate is much smaller than that in the Pro group, which again indicates that the carrier ZnO@Pectin can effectively reduce the migration of Pro in the soil (Figure 5H).

Pesticide residues are trace amounts of pesticide prototypes that remain in agricultural products without being decomposed for a period of time after pesticide use. They are the inevitable result of pesticide application. If the maximum residue limit is exceeded, toxicity will occur. In the previous article, the relationship between the wettability of nanoformulations and pesticide residues was mentioned. The residue of PZP on cherry tomatoes was evaluated by measuring the content of Pro in the eluate after rinsing the fruits (Figure 5I). The results showed that the Pro contents of the eluates obtained by washing fruits in different ways in daily life, such as rinsing and soaking, were measured and calculated to be 444.31 μg and 443.38 μg in the PZP group, and 436.91 μg and 453.24 μg in the Pro group (Figure 5J). The corresponding maximum pesticide residues were 1.33 mg/kg, 1.21 mg/kg, 1.57 mg/kg, and 1.35 mg/kg (Figure 5K), which were all lower than the maximum allowable residue of 2 mg/kg stipulated in Solanaceae vegetables. In addition, the acceptable daily intake (ADI) of prochloraz is 0.01 mg/kg, which means that an adult weighing 60 kg will not cause harm to health if the daily intake is less than 0.6 mg. According to the maximum residue calculation obtained from the experiment, the prochloraz residue must be consumed in more than 0.5 kg of tomatoes before it will harm health. In real life, more prochloraz is removed through washing. In summary, although there are pesticide residues in PZP and Pro, they are all within the normal range. The nanocarriers increase the adhesion without increasing the residue of Pro.

## 4. Conclusions

This research unveils PZP, a novel dual-response nanopesticide engineered to combat *Sclerotinia* infections effectively. PZP consists of zinc oxide (ZnO) loaded with the fungicide prochloraz and encapsulated with pectin. PZP ingeniously reacts to the presence of oxalic acid and fungal cell wall-degrading enzymes, triggering the smart release of its active compounds. Impressively, even at reduced doses, PZP outperforms conventional PMBD in terms of antimicrobial effectiveness, underscoring the potency of its innovative responsive release system. Field trials affirm PZP’s efficacy in hindering *Sclerotinia* progression, hinting at its potential as a transformative therapeutic solution. Safety evaluations confirm PZP’s gentle touch on wheat, showing negligible impact on seed germination and seedling development, and mitigating the phytotoxicity typically associated with prochloraz. Environmental assessments highlight PZP’s prolonged stability without ecological harm, emphasizing its eco-friendly credentials. Furthermore, residual analysis and wettability tests indicate that while the nanocarrier enhances overall system wettability, it does so without significantly escalating residue levels, thereby managing risks inherent to nanotech applications. In conclusion, PZP emerges as a promising disease management strategy, seamlessly integrating safety, environmental sustainability, and robust disease control into a single innovative package.

## Data Availability

Data is contained within the article or Appendix A.

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
