# Peer review of "pH and Pectinase Dual-Responsive Zinc Oxide Core-Shell Nanopesticide: Efficient Control of Sclerotinia Disease and Reduction of Environmental Risks"

_nanomaterials, 2024, doi:10.3390/nano14242022_

Round 1
Reviewer 1 Report
Comments and Suggestions for Authors
The article deals to the development of a novel approach for protecting plants against the fungus Sclerotinia sclerotiorum. The authors propose the use of a novel hybrid material consisting of commercially available prochloraz fungicide, polysaccharide pectin and synthesized zinc oxide particles as a potent fungicidal agent. Zinc oxide particles act as containers for prochloraz, while pectin serves as a protective shell for the obtained particles. The authors demonstrate that prochloraz release from this material occurs in the presence of Sclerotinia sclerotiorum-specific metabolic products, including oxalic acid and pectinase.
The paper's topic aligns with the scope of Nanomaterials, and its conclusions are certainly of interest to the journal's readership. The paper is well-structured, presents material in a consistent manner, and provides justified conclusions supported by the results. This study represents progress in nanopesticide research, making it suitable for publication in Nanomaterials after addressing the following comments:
1. The authors describe the antibacterial activity of the obtained material against Sclerotinia sclerotiorum. Antibacterial activity refers to activity against bacteria, which are prokaryotes, whereas Sclerotinia sclerotiorum belongs to the eukaryotic kingdom of fungi. Therefore, the term “antibacterial activity” should be replaced with “fungicidal activity” throughout the manuscript.
2. In Section 2.2.1 (Page 3, Line 100), it is stated that zinc oxide was synthesized using a modified method from reference [43]. However, reference [43] uses commercial ZnO, and no synthesis method is described there.
3. The manuscript lacks confirmation of the formation of a zinc oxide phase during the synthesis described. X-ray diffraction analysis of the obtained particles should be provided.
4. The authors state that the obtained zinc oxide is nanocrystalline; however, TEM data indicate that its particle size is slightly over 200 nm.
5. The nitrogen adsorption–desorption isotherm for zinc oxide is presented, but the specific surface area is not provided. What type of isotherm do the authors classify it as, according to the IUPAC classification? Can the pore size distribution be calculated from the isotherm?
6. The first sentence of the abstract states that plants release a large amount of oxalic acid and pectinase. This is inaccurate, as these are released by the fungus Sclerotinia sclerotiorum, not the infected plants.
7. The abbreviations for the samples (Pro, PZP, PSC) should be introduced not only in the abstract but also in the experimental section.
8. Please specify the sample sizes used in Section 2.15.
9. Please indicate the purity of the reagents used in the study.10. Page 10, Line 332: replace "scatterdd" with "scattered."
11. Please verify the caption for Figure 3; (F) does not appear to be a photograph.
Considering these remarks and questions, I recommend a major revision of the manuscript.
Reviewer 2 Report
Comments and Suggestions for Authors
Specific Comments:
Provide a detailed discussion or schematic representation of the dual-responsive mechanism.
There is a need to include data about how oxalic acid and pectinase influence ZnO and pectin degradation.
You may investigate whether the enhanced antibacterial effect is due to better bioavailability, targeted release, or prolonged efficacy of PZP.
Is it possible to limit negative impact on non-target organisms.
Compare the residue levels of PZP with established residue limits to reassure regulatory compliance.
Explain how to improve leaf wettability that may enhance pesticide coverage, adhesion, and absorption
It is possible of controlling Sclerotinia sclerotiorum under real-world conditions.
Discuss the feasibility of large-scale production and potential cost-effectiveness compared to traditional pesticides.
Discuss current regulatory concerns, possible toxicity concerns, and significance of extensive field trials to pave the way for practical implementation.
Round 2
Reviewer 1 Report
Comments and Suggestions for Authors
The authors have addressed most of my comments, while the following issue remain unanswered:
Specifically, TEM data confirm that the size of the particles is about 200-400 nm, and so they cannot be referred to as nanoparticles. However, these particles seem to be composed of much smaller crystallites. To confirm the nanocrystalline nature of the materials synthesized, the authors should estimate the crystallite sizes from the XRD data.
English should be improved throughout the paper.
